# Pharmaceutical Evaluation of Levofloxacin Orally Disintegrating Tablet Formulation Using Low Frequency Raman Spectroscopy

**DOI:** 10.3390/pharmaceutics15082041

**Published:** 2023-07-29

**Authors:** Yoshihisa Yamamoto, Mizuho Kajita, Yutaro Hirose, Naoki Shimada, Toshiro Fukami, Tatsuo Koide

**Affiliations:** 1Faculty of Pharmaceutical Sciences, Teikyo Heisei University, 4-21-2 Nakano, Nakano-ku, Tokyo 164-8530, Japan; 2Bio & Healthcare Division, HORIBA Ltd., 2 Miyanohigashi-cho, Kisshoin, Minami-ku, Kyoto 601-8510, Japan; mizuho.kajita@horiba.com (M.K.); yutaro.hirose@horiba.com (Y.H.); 3Department of Molecular Pharmaceutics, Meiji Pharmaceutical University, 2-522-1 Noshio, Kiyose-shi, Tokyo 204-8588, Japan; m226227@std.my-pharm.ac.jp; 4Division of Drugs, National Institute of Health Sciences, 3-25-26 Tonomachi, Kawasaki-ku, Kawasaki 210-9501, Japan; koide@nihs.go.jp

**Keywords:** levofloxacin, orally disintegrating tablet, crystalline form, low frequency Raman spectroscopy

## Abstract

We evaluated the pharmaceutical properties of levofloxacin (LV) in the form of an orally disintegrating tablet (LV*_ODT_*) to find a new usefulness of low frequency (LF) Raman spectroscopy. LV*_ODT_* contained dispersed granules with diameters in the order of several hundred micrometers, which were composed of the active pharmaceutical ingredient (API), as confirmed by infrared (IR) microspectroscopy. On the contrary, the API and inactive pharmaceutical ingredients (non-APIs) were homogeneously distributed in LV tablet (LV*_T_*) formulations. Microscopic IR spectroscopy and thermal analyses showed that LV*_ODT_* and LV*_T_* contained the API in different crystalline forms or environment around the API each other. Furthermore, powder X-ray diffraction showed that LV*_T_* contained a hemihydrate of the API, while LV*_ODT_* showed a partial transition to the monohydrate form. This result was confirmed by microscopic LF Raman spectroscopy. Moreover, this method confirmed the presence of thin layers coating the outer edges of the granules that contained the API. Spectra obtained from these thin layers indicated the presence of titanium dioxide, suggesting that the layers coexisted with a polymer that masks the bitterness of API. The microscopic LF Raman spectroscopy results in this study indicated new applications of this method in pharmaceutical science.

## 1. Introduction

Microscopic imaging systems that include X-ray fluorescence, infrared (IR) radiation, near-IR radiation, terahertz technology, Raman spectroscopy, and numerous other spectroscopic techniques have recently become valuable analytical tools in pharmaceutical design and quality control. Microspectroscopic imaging systems are able to consecutively determine the spectra of ultrasmall pixels in a plane of the sample, analyze the spectral data, and produce images with the results combined to obtain two-dimensional chemoinformatic images. Distributions of the active pharmaceutical ingredients (APIs) and excipients in pharmaceutical formulations have been successfully imaged using microscopic IR spectroscopy [1,2,3]. API conversions to magnesium hydroxide as a result of moisture absorption on the surface of magnesium oxide tablets have also been successfully visually evaluated [4].

Raman spectroscopy involves the detection of scattered light with changes in frequency obtained by irradiating a sample with a laser. Since this method detects a peak at a specific Raman shift for each chemical bond, the molecular structure can be estimated in a manner similar to IR spectroscopy. Low frequency (LF) Raman spectroscopy focuses on Raman shifts < 500 cm^−1^ and recently attracted attention as a technique for detecting differences in crystal types unlike with conventional Raman spectroscopy [5,6,7,8,9].

Levofloxacin (LV), a new quinolone antibacterial agent, is available in Japan in more than 20 tablet formulations (LV*_T_*) and one orally disintegrating tablet (LV*_ODT_*). Because LV has a strong bitterness, LV*_ODT_* was developed to disintegrate quickly in the oral cavity, making it difficult for the patient to perceive the bitterness.

Herein, we evaluated the formulation characteristics of LV*_ODT_* by first visually evaluating the distribution of LV and the additives in LV*_T_* and LV*_ODT_* using microscopic infrared spectroscopy. The results obtained with this method were then verified in detail using thermal analyses and powder X-ray diffraction (PXRD). Furthermore, to clarify the effectiveness of LF Raman spectroscopy, we evaluated whether the information obtained could be confirmed by conventional methods.

## 2. Materials and Methods

### 2.1. Materials

LV*_ODT_* (250 mg “Towa”, lot. B0048) was purchased from Towa Pharmaceutical Co., Ltd. (Osaka, Japan). Cravat tablets (250 mg, lot. QUA1173, Daiichi Sankyo Co., Ltd., Tokyo, Japan; LV*_T_CRABIT_*), 250 mg levofloxacin tablets “Nipro” (lot. 20L031, Nipro, Osaka, Japan; LV*_T_NIPRO_*), 250 mg levofloxacin tablets “Sawai” (lot. 420502, Sawai Pharmaceutical Co., Ltd., Osaka, Japan; LV*_T_SAWAI_*), and 250 mg levofloxacin tablets “Towa” (lot. B0023, Towa Pharmaceutical Co., Ltd., Osaka, Japan; LV*_T_TOWA_*) were analyzed. The non-API formulations for all tablets are shown in Table 1 [10,11,12,13].

LV was purchased from Tokyo Chemical Industry Co., Ltd. (L0193, Tokyo, Japan) and Dadipharm (Lot. HBW211015-7, Pingxiang, China). Both were commercially available levofloxacin hemihydrate (LV_0.5_) reagents, which is stable under ambient conditions. Microcrystalline cellulose (MCC; CEOLUS^®^ UF-702) was purchased from Asahi Kasei Chemicals (Tokyo, Japan).

### 2.2. Preparation of Levofloxacin Mono Hydrate (LV_1.0_)

A suspended solution was obtained by mixing 7.01 g of LV_0.5_ in 42 mL of a 50% ethanol aqueous solution and stirring for 12 h at 40 °C. After filtration, the filtrate was left to stand at ambient conditions [14].

### 2.3. Microscopic IR Spectroscopy Measurements

The LV*_ODT_* and LV*_T_* formulation surfaces were carefully thinly scraped and used for subsequent measurements. A NICOLET iN10 IR microscope (Thermo Scientific, Yokohama, Japan) was used to collect the IR spectra of the tablet formulations using the reflection method. The background was measured on gold, and sample scans were recorded at a spectral resolution of 8 cm^−1^ with 16 scans in a range of 4000–675 cm^−1^. Data were analyzed using the OMNIC Picta chemical imaging software (PN:81032530, Thermo Scientific). Mapping was performed using the peak area from 1700 cm^−1^ to 1800 cm^−1^ (PA*_1700–1800_*, Map 1) and correlated to the indices of the standard spectrum of MCC (CR*_MCC_*, Map 2).

The mapping conditions included an aperture size of 100 µm × 100 µm; step size, 100 µm; and measurement area, 2000 µm × 2000 µm (number of measurement points: 20 point × 20 point).

The standard IR spectra of LV and MCC were measured in “spectrum mode” under the same conditions described above.

### 2.4. Separation of LV_ODT_ and LV_T_ Formulations by Particle Size

The LV*_ODT_* and LV*_T_* formulations were lightly crushed with a mortar and pestle and sorted into large (d ≥ 355 µm), medium particles (75 µm ≤ d < 355 µm), and small (d < 75 µm) particles by sieving. Differential scanning calorimetry (DSC), thermogravimetric-differential analysis (TG-DTA), and PXRD measurements were performed for each particle.

### 2.5. Thermal Analyses

The DSC and TG-DTA measurements were performed by differential scanning calorimetry (Thermo plus EVO2, DSCvesta, Rigaku Corp., Tokyo, Japan) and a thermogravimetry (Thermo plus EVO2, TG-DTA8122, Rigaku Corp.). A 5 mg sample was placed in an aluminum crucible and heated from 20 °C to 490 °C at a rate of 10 °C/min under a nitrogen atmosphere. The reference material was air.

### 2.6. PXRD

PXRD analysis was performed using a MiniFlex600 benchtop X-ray diffractometer (Rigaku Corp.) operated at 40 kV and 15 mA with Cu Kα. The data were acquired at a scanning speed of 20.0°/min, diffraction angles (2θ) in the range of 5.0–35.0°, and sampling width of 0.02°.

### 2.7. Microscopic Raman Spectroscopy (Conventional and LF)

The surface of LV*_ODT_* and LV*_T_* formulations was carefully scraped and used for measurement. A LabRAM HR Evolution Raman microscope (Horiba, Kyoto, Japan) was used to collect the Raman spectra of the tablet formulations. The Raman microscopy system consisted of a 532 nm laser, 100 mW power source, an electron multiplying charged-coupled device camera (SynapseEM, Horiba), and a microscope with a 50× objective lens (numerical aperture = 0.75, Olympus, Tokyo, Japan). Conventional Raman measurements used a 150 gr/mm grating and measured the spectrum from 500 cm^−1^ to 4000 cm^−1^. LF Raman measurements used a 1800 gr/mm grating and LF Raman unit (ULF-532, Horiba) to measure spectra from 10 cm^−1^ to 400 cm^−1^. Data were analyzed using Labspec6 (Horiba).

The mapping conditions used were an exposure time of 1 s; neutral density (ND) Filter, 3.2%; hole, 200 µm; and step size, 10 µm × 10 µm.

Raman spectra of LV_0.5_ were obtained using the conditions: exposure time, 10 s × 2; ND Filter, 5%; and hole, 200 µm.

## 3. Results and Discussion

### 3.1. Visual Evaluation of LV_ODT_ and LV_T_ Formulations by Microscopic IR Spectroscopy

The distributions of LV, the API in LV*_ODT_* and LV*_T_* formulations, were visually evaluated by microscopic IR spectroscopy. Figure 1 shows the standard IR spectra of LV and MCC. The LV standard (LV*_ST_*) showed a characteristic peak at 3200–3500 cm^−1^ derived from the hydroxyl groups of the carboxylic acids and crystalline water. In addition, a sharp peak that was observed between 1700 cm^−1^ and 1800 cm^−1^ for LV*_ST_* derived from carbonyl group was completely absent in the MCC spectrum. Therefore, we determined that the distribution of LV could be visually evaluated by creating a mapping image using the values within the peak area from 1700 cm^−1^ to 1800 cm^−1^ (PA*_1700–1800_*) as an index.

Figure 2a shows the microscopic image, and Figure 2b,c show the mapping images of the LV*_ODT_*, respectively. To confirm the distribution of API in LV*_ODT_*, we created a mapping image using the PA*_1700–1800_* values as an index (Map 1, Figure 2b). High- and low-area regions (red and blue, respectively) were observed in Map 1 of LV*_ODT_* and were clearly distinguishable from each other (Figure 2b). Furthermore, the distributions of the red region and that of the white granules in the microscopic image (Figure 2a,b) matched perfectly. We also created a mapping image using the correlation to the MCC standard spectrum (Map 2; Figure 2c). In Map 2, a clear separation of the highly correlated (red) and low-correlated (blue) regions was obtained, similar to Map 1, but their distributions was in complete contrast to that of Map 1. Furthermore, the distribution of the red regions in Map 2 matched perfectly with the areas other than the white granules in the microscopic images (Figure 2a,c). The spectrum obtained from the red regions of Map 1 was similar to the standard spectrum of LV (Figure 2d), and the spectrum obtained from the blue regions was similar to the standard spectrum of MCC (Figure 2e). These results suggest that the structure of LV*_ODT_* was such that the API-containing granules with particle sizes of several hundred micrometers are distributed within the formulation.

However, in LV*_T_TOWA_*, although red and blue region distributions were observed in both Map 1 and Map 2, there was no correlation with the microscopic images (Figure 3a–c). In both spectra obtained from the pixels in the red and blue regions of Map 1 (Figure 3b), a clear peak in the range of 1700–1800 cm^−1^ was observed, which was different from that of LV*_ODT_* (Figure 3d,e). These results suggest that the LV*_T_TOWA_* spectra contained information on both the API and non-APIs at all measurement points and that LV and non-API particles were uniformly distributed; however, slight differences in the ratios of API and non-API particles at different measurement points were seen. These results were also observed for the other LV*_T_* formulations used in this study (LV*_T_CRAVIT_*, LV*_T_NIPRO_*, and LV*_T_SAWAI_*; Appendix A, respectively).

Comparing the spectra obtained from the red regions in the mapping images (Map 1) of LV*_ODT_* and LV*_T_TOWA_* revealed differences in the peak shapes in the range of 3200 cm^−1^–3500 cm^−1^ (Figure 2d and Figure 3d). In particular, a clear peak was observed for LV*_T_TOWA_* at approximately 3250 cm^−1^, which was also seen in LV*_ST_* (Figure 1) but not in LV*_ODT_*. This suggests differences in the crystal forms or environments around the API between LV*_ODT_* and LV*_ST_* or LV*_T_TOWA_*.

### 3.2. DSC and TG-DTA Measurements in Lightly Crushed LV_ODT_ and LV_T_ Formulations

Microscopic IR spectroscopy revealed that in LV*_ODT_*, the API-containing granules were well distributed within the formulation. Therefore, we attempted to physically separate the regions of granules containing the API from other than granules containing non-APIs to examine the crystalline forms of the API granules in greater detail. For LV*_ST_*, an endothermic peak was observed at approximately 70 °C. The accompanying mass reduction (Appendix A) suggested that this peak was due to the release of crystalline water. An endothermic peak without mass reduction was observed at approximately 235 °C (Figure 4a,b and Appendix A), indicating that it was caused by the decomposition of the API [9,10,11,12].

For the large particles of LV*_ODT_*, an endothermic peak was observed at approximately 220 °C, which was likely due to the decomposition of API and was approximately 10 °C lower than that of LV*_ST_*; the shape of the peak was also apparently different (Figure 4a). These differences suggest that the crystalline form or the environment around the API in LV*_ODT_* was different from those of LV*_ST_* determined by thermal analyses. Furthermore, this endothermic peak was observed for large and medium particles but not for small particles (Figure 4a). The enthalpies of melting calculated from the peak areas for the large and medium particles were 30.4 and 16.1 J/g, respectively, and unquantifiable for the small particles. These results suggest the presence of the API in large and medium particles and that the granules with diameters of several hundred micrometers observed in the microscopic images were contained in these particles. In this formulation, endothermic peaks at approximately 170 °C and 290 °C were observed, which were absent in the LV*_ST_* data (Figure 4a). The peak at 170 °C was not accompanied by a mass reduction (Appendix A), indicating an endothermic reaction associated with the melting of mannitol, which is a non-API in LV*_ODT_* (Table 1). In contrast, the peak at 290 °C was accompanied by a mass reduction (Figure 4a and Appendix A), suggesting an endothermic reaction associated with the thermal decomposition of non-APIs containing MCC. These endothermic peaks increased with decreasing particle diameters (Figure 4a and Appendix A), suggesting that mannitol and MCC were originally distributed outside of the granules and were present in higher concentrations around particles with smaller diameters due to milling.

For LV*_T_TOWA_*, the onset temperatures of the peaks associated with the release of crystalline water and the decomposition of the API were almost identical from those of LV*_ST_*, with only small differences in the enthalpies of the API for each particle size (32.8, 49.5, and 45.0 J/g for large, medium, and small particles, respectively) (Figure 4b and Appendix A). These results suggest that the API was homogeneously distributed in LV*_T_TOWA_* regardless of the particle size and existed in the same crystalline form as LV*_ST_*, which was consistent with the results obtained by microscopic IR spectroscopy.

### 3.3. PXRD of the LV_ODT_ and LV_T_ Formulations

Since results of previous measurements (Section 3.1 and Section 3.2) suggested that the crystalline form of the API or the environment around the API differed between LV*_ODT_* and LV*_T_TOWA_*, we used PXRD to obtain thermal measurements of particles in each diameter class (large, medium, and small particles). The spectra of LV*_ODT_* did not match those of LV*_ST_*, with the peak intensities varying with diameter (Figure 5a). Combined with the thermal analysis results, this suggested that the peak at approximately 7.5°, which became smaller as the particle size decreased, reflected the presence of the API, while the peak at approximately 24°, which became larger as the particle size decreased, reflected the presence of some non-APIs. The absence of the diffraction peak at approximately 7.5° in LV*_ST_* confirmed that the crystal form and the surrounding environment of the API in LV*_ODT_* were different from those of LV*_ST_*. In contrast, the spectra of LV*_T_TOWA_* were almost identical to those of LV*_ST_* at all particle diameters, confirming that the crystal form of the API in LV*_T_TOWA_* was consistent with that of LV*_ST_*, and that the API was uniformly distributed (Figure 5b). According to the interview forms for each formulation used in this study [9,10,11,12], the API contained in LV*_T_TOWA_* was LV_0.5_, suggesting that LV*_ST_* was also LV_0.5_. PXRD measurements of experimentally prepared LV_0.5_ and LV_1.0_ were performed and compared with the spectra of the LV*_ODT_* particles at different diameters, which revealed diffraction peaks at similar angles of incidence as those of the LV_1.0_ for the large and medium particles (Figure 5a). A specific peak observed for the large and medium particles of LV_0.5_ at approximately 6° (Figure 5a) suggested that the part of the API used in LV*_ODT_* was a transition state from LV_0.5_ to LV_1.0_.

### 3.4. Visual Evaluation of LV_ODT_ and LV_T_ Formulations by Conventional and LF Raman Microspectroscopy

Next, we tested whether microscopic Raman spectroscopy could provide additional useful information.

Figure 6 shows a mapping image drawn using conventional Raman spectroscopy data of LV*_ODT_* and LV*_T_TOWA_*. For LV*_ODT_*, characteristic spectra were obtained from the regions of white granules and brown nongranular regions in the microscopic images (Figure 6a). Figure 6b shows the average spectra obtained from part of the nongranular region and the granular region (areas 1 and 2 of Figure 6a, respectively). Specific peaks were detected at approximately 1600 cm^−1^ and 2900 cm^−1^ for the regions with and without granules (areas 2 and 1), respectively (Figure 6b); Raman images were drawn in red and blue using the intensity of each peak as an indicator. The regions of high intensity at 1600 and 2900 cm^−1^ perfectly coincided with the regions with and without granules in the microscopic image, respectively (Figure 6a). The average Raman spectra obtained from the granule regions (area 2) were almost identical to those of LV*_ST_* (Figure 6b). However, there was no difference in the spectra of the LV*_T_TOWA_* for various sites, with all of the spectra being almost identical to those of LV*_ST_* (Figure 6c,d). The results obtained by conventional Raman spectroscopy were similar to those obtained by microscopic IR spectroscopy, with no new information obtained using this technique.

We then performed similar measurements using microscopic LF Raman spectroscopy. For LV*_ODT_*, characteristic peaks were obtained from a thin layer (approximately 20 µm) at the outer edge of the granules in addition to the peaks form the white granules and nongranular regions that were also observed in the conventional Raman spectra (Figure 7a). The average spectra obtained from the nongranular region (area 1), granules (area 2), and the thin layer at the outer edge of the granules (area 3) had specific peaks at 55, 20, and 150 cm^−1^, respectively (Figure 7b); Raman imaging was, therefore, performed using the intensities of these wavenumbers as indicators, marked in green, red, and blue, respectively. The red region corresponded to the white granules in the microscopic image and the blue layer, which was approximately 20 µm thick, was found on the outer edge of the red region (Figure 7a). The API-derived spectrum from the red region was similar but not a perfect match to that of LV*_ST_* (Figure 7b).

The LF Raman spectra of experimentally prepared LV_1.0_ included peaks at approximately 25 and 40 cm^−1^. The peak at 40 cm^−1^ was specific to LV_1.0_ (Appendix A). Moreover, the average spectrum obtained from area 2 included a peak at 52 cm^−1^ that appeared to be derived from LV_0.5_ (Figure 7b), strongly suggesting that part of the API used in LV*_ODT_* transition from the LV_0.5_ to the LV_1.0_, as observed using other methods. In contrast, the shape of the peak at 25 cm^−1^ (Appendix A) differed from that of the spectrum obtained from the red region (Figure 7b), which was attributed to differences in instrument resolutions.

Furthermore, the spectrum of the blue layer indicated not only the presence of API due to the peak below 50 cm^−1^ being consistent with that of the red region, but also the strong characteristic peak around 150 cm^−1^. This striking peak was assigned with that of titanium dioxide, the non-API mainly used as mostly sunscreen in tablet coatings (Figure 7b and Appendix A). It is notable that the peak was recognized solely in the measurement of LF region, since the peaks of titanium dioxide could be overlapped with other ingredients in conventional region. LV itself is a very bitter compound, making it necessary to mask this bitterness for easy administration as an orally disintegrating tablet. Aminoalkyl methacrylate copolymer E is used as a bitterness-masking agent along with this drug; the granules would be coated with this polymer. The titanium dioxide is a common pharmaceutical ingredient widely used with various polymers for film coating. Therefore, titanium dioxide is likely to coexist with this polymer and could visualize the coating layer in LF Raman measurement.

We showed that the shape of the endothermic peak due to LV melting observed in LV*_ODT_* during the thermal measurements was significantly different from those of LV*_ST_* and LV*_T_TOWA_* (Figure 4a). This may have been due to the heat transfer to API being not as smooth as that of LV*_T_TOWA_* because of the coating of the granules. In contrast, no obvious differences were observed between the spectra from any of the LV*_T_TOWA_* sites, with all spectra consistent with LV*_ST_* (Figure 7c,d). These results for LV*_T_TOWA_* by conventional and LF Raman spectroscopy (Figure 6c,d and Figure 7c,d) were also observed for other LV*_T_* formulations (LV*_T_CRAVIT_*, LV*_T_NIPRO_*, and LV*_T_SAWAI_*). This demonstrated that the LV*_T_* formulations contain LV_0.5_, with a uniform distribution of the API and non-APIs, which was consistent with the results of previous studies using other methods.

The microscopic LF Raman spectroscopy used in this study allowed us to discriminate between the crystalline forms of API in LV*_ODT_* and LV*_T_*, as well as visually analyze the distribution of the approximate 20 µm thickness film coating that covers the outer surface of the granules in LV*_ODT_*. The results showed that microscopic LF Raman spectroscopy can detect changes in the physical properties of generic formulations, which represents the novel effectiveness of this method in the field of pharmaceutical science.

## 4. Conclusions

In this study, we verified the distribution of the API and non-API in LV*_T_* formulations and LV*_ODT_* by microscopic IR spectroscopy, thermal analysis, PXRD, and microscopic Raman spectroscopy. The results showed that fine particles of the API and non-APIs were uniformly distributed in the formulation of LV*_T_* formulations, whereas granules with a diameter of several hundred micrometers were distributed in the formulation of LV*_ODT_* containing the API. Moreover, we successfully imaged the thin coating layer around the API granules in LV*_ODT_* by micro LF Raman spectroscopy. This information could not be obtained via other measurement methods, indicating the importance of LF Raman spectroscopy. Although there are few reported cases of LF Raman imaging, as shown in this study, this method can recognize differences in hydration. Furthermore, this method can identify the special structure of OD tablets, which is useful for identifying the cause of defective tablets and important for controlling the design and manufacturing process.

Thus, we believe that it is very significant that the LF-Raman technique enabled us to visually evaluate areas that could not be evaluated by IR or conventional Raman imaging measurements. This result by LF Raman was the most significant topic in this study. It is considered that the combination of results of LF Raman spectroscopy with that of various other methods will enable more reliable evaluation of drug formulations in the future.

## Figures and Tables

**Figure 1 pharmaceutics-15-02041-f001:**
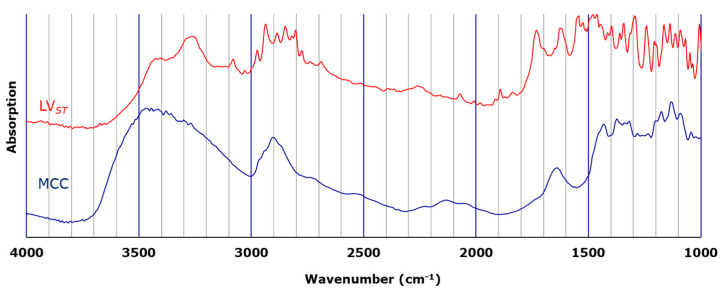
Standard infrared (IR) spectra of standard levofloxacin (LV*_ST_*; red) and microcrystalline cellulose (MCC; blue).

**Figure 2 pharmaceutics-15-02041-f002:**
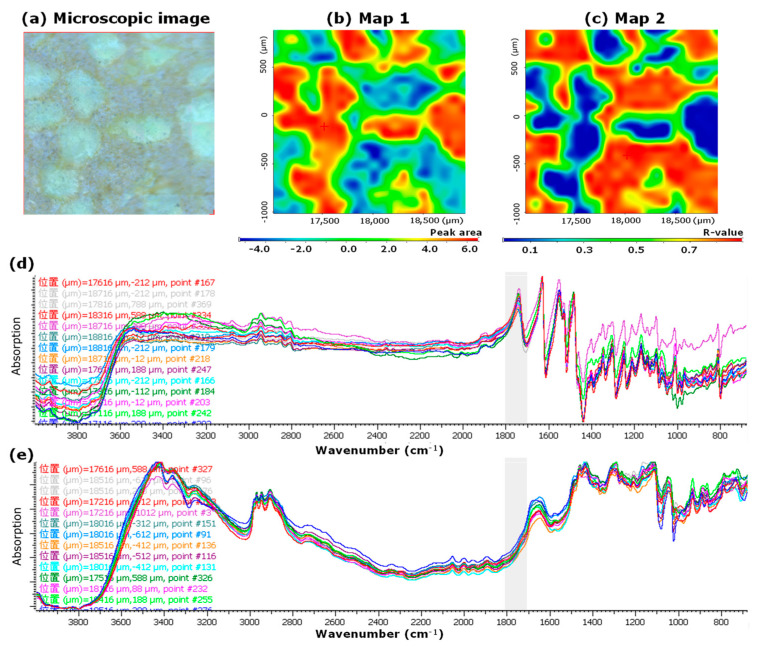
Microscopic and mapping image of LV*_ODT_* by microscopic IR spectroscopic method (2000 μm × 2000 μm). Microscopic image (**a**), Map 1; mapping image obtained from the 1700 to 1800 cm^−1^ peak area (PA_1700–1800_) of each spectrum. The red and blue regions indicate high and low peak areas, respectively. The measurement area was 2000 µm × 2000 µm (**b**), Map 2; mapping image obtained from the correlation to standard microcrystalline cellulose (MCC) spectrum (CR*_MCC_*). The red and blue regions indicate high and low correlations, respectively. The measurement area was 2000 µm × 2000 µm (**c**), IR spectra obtained from the measurement points in the red (**d**) and blue regions (**e**) of Map 1. “位置” in (**d**,**e**) means the position.

**Figure 3 pharmaceutics-15-02041-f003:**
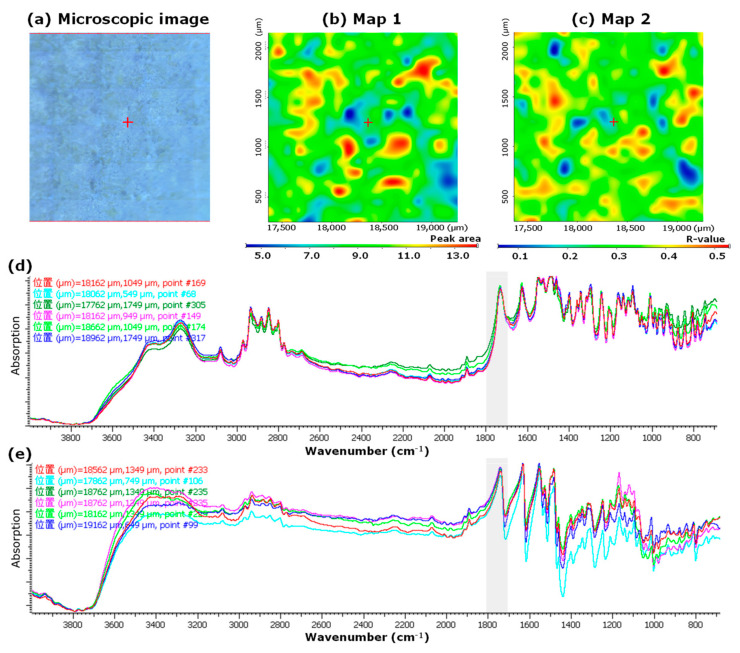
Microscopic and mapping images of the LV tablet formulation LV*_T_TOWA_* by microscopic IR spectroscopy (2000 μm × 2000 μm). Microscopic image (**a**), Map 1; mapping image obtained from the PA*_1700–1800_* of each spectrum. The red and blue regions indicate the high and low peak areas, respectively. The measurement area was 2000 µm × 2000 µm (**b**), Map 2; mapping image obtained from the CR*_MCC_* of each spectrum. The red and blue regions indicate high and low correlations, respectively. The measurement area was 2000 µm × 2000 µm (**c**), IR spectra obtained from the measurement points in the red (**d**) and blue regions (**e**) of Map 1. “位置” in (**d**,**e**) means the position.

**Figure 4 pharmaceutics-15-02041-f004:**
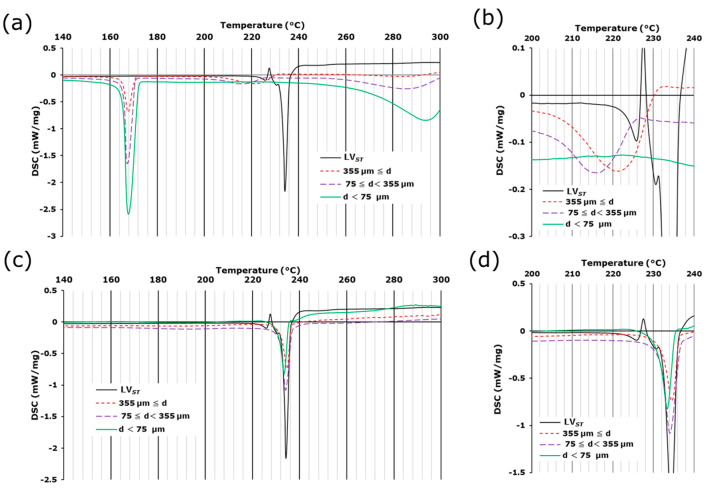
DSC curves of lightly crushed LV*_ODT_* (**a**,**b**) and LV*_T_TOWA_* (**c**,**d**) with several particle sizes. Standard levofloxacin is labeled as LV*_ST_*. (**a**,**c**): 140–300 °C, (**b**,**d**): 200–240 °C.

**Figure 5 pharmaceutics-15-02041-f005:**
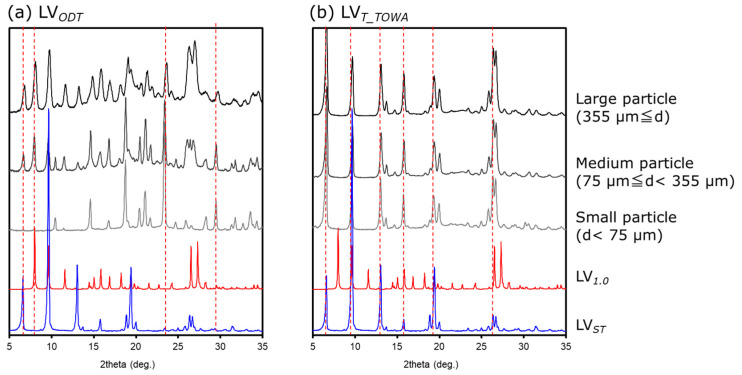
Diffraction patterns of crushed (**a**) LV*_ODT_* and (**b**) LV*_T_TOWA_* for several particle sizes.

**Figure 6 pharmaceutics-15-02041-f006:**
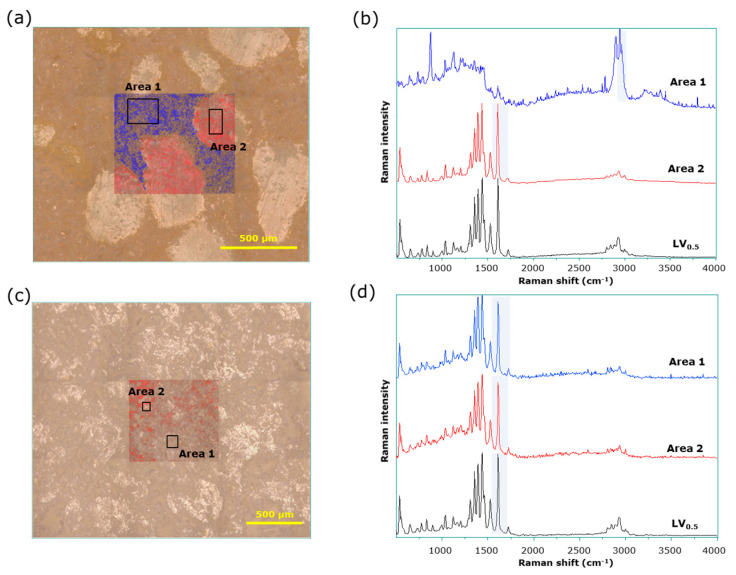
Conventional Raman spectroscopy of LV*_ODT_* and LV*_T_TOWA_*. (**a**) Microscopic and mapping images of LV*_ODT_*; (**b**) average Raman spectra obtained from blue area (area 1) and red area (area 2) of mapping image; (**c**) microscopic and mapping images of LV*_T_TOWA_*; (**d**) average Raman spectra obtained from area 1 and area 2 of mapping image.

**Figure 7 pharmaceutics-15-02041-f007:**
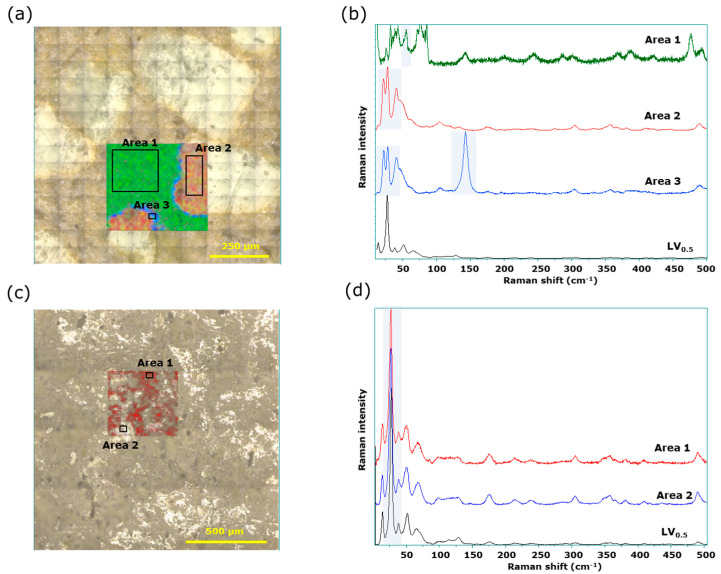
Microscopic LF Raman spectroscopy of LV*_ODT_* and LV*_T_TOWA_*. (**a**) Microscopic and mapping images of LV*_ODT_*; (**b**) average Raman spectra obtained from the green area (area 1), red area (area 2), and blue area (area 3) of mapping image; (**c**) microscopic and mapping images of LV*_T_TOWA_*; (**d**) average Raman spectra obtained from area 1 and area 2 of mapping image.

**Table 1 pharmaceutics-15-02041-t001:** Nonactive pharmaceutical ingredients of LV*_ODT_* and LV*_T_* formulations.

Formulations	Non-APIs *
LV*_ODT_*	MCC ** Carmellose sodium, Hydroxypropyl cellulose, Sucralose, Aminoalkyl methacrylate copolymer E, Talc, Titanium dioxide, Yellow ferric oxide, D-Mannitol, MCC **, Light anhydrous silicic acid, Fragrance, Magnesium stearate, other 3 components
LV*_T_CRAVIT_*	MCC **, Carmellose, Hydroxypropyl cellulose, Stearyl sodium fumarate, Hypromellose, Titanium dioxide, Talc, Macrogol 6000, Yellow ferric oxide, Carnauba wax
LV*_T_NIPRO_*	MCC **, Hydroxypropyl cellulose, Carmellose, Stearyl sodium fumarate, Hypromellose, Macrogol, Talc, Titanium dioxide, Yellow ferric oxide, Carnauba wax
LV*_T_SAWAI_*	Carnauba wax, Carmellose, MCC **, Titanium dioxide, Ferric oxide, Talc, Hydroxypropyl cellulose, Hypromellose, Stearyl sodium fumarate, Macrogol 6000
LV*_T_TOWA_*	MCC **, Carmellose, Hydroxypropyl cellulose, Cros-carmellose sodium, Magnesium stearate, Hypromellose, Macrogol 6000, Talc, Titanium dioxide, Yellow ferric oxide

***** These inactive pharmaceutical ingredients (non-APIs) have been cited from manufacturer’s forms [10,11,12,13]. ** MCC: Microcrystalline cellulose

## Data Availability

All data are presented within the manuscript and Appendix A.

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
