# Peer review of "Pharmaceutical Evaluation of Levofloxacin Orally Disintegrating Tablet Formulation Using Low Frequency Raman Spectroscopy"

_pharmaceutics, 2023, doi:10.3390/pharmaceutics15082041_

Round 1

Reviewer 1 Report

Review of Pharmaceutics "Pharmaceutical evaluation of Levofloxacin orally disintegrating tablet formulation using low frequency Raman spectroscopy"  by Yoshihisa Yamamoto et al.

This is a very thorough piece of work in which a number of tableted formulations containing levofloxacin are examined using both infrared and Raman microscopy and these are related to bulk methods such as DSC and XRPD.  The tablets are crushed to facilitate some of the measurements; again, there is careful assessment of the varied samples.

Ultimately it is found that low frequency Raman microscopy offers the ability to detect the coating about the particles of API – and that is really impossible to detect effectively with any other method.  In this sense the power of low frequency Raman spectroscopy – or in this case microscopy is showcased.

I have a couple of comments:

1.   I do not think there is a need for section 4 as setion 3 combines results and discussion

2.      I thought the conclusions were a little terse.  The study does carefully investigate a number of tableted forms of levofloxacin – perhaps that could also be mentioned.

Minor revisions with no need for re-review

Author Response

1.  I do not think there is a need for section 4 as setion 3 combines results and discussion

Answer: Thank you for pointing that out. We have deleted the phrase "4. Discussion" and have revised the “5. Conclusion" to "4. Conclusion" (L. 353 in revised manuscript).

2.   I thought the conclusions were a little terse.  The study does carefully investigate a number of tableted forms of levofloxacin – perhaps that could also be mentioned.

Answer: Other reviewers have pointed out that the conclusions are too brief. We also added the point that this study clarified the structural differences between LVODT and LVT formulations (L. 354-366 in revised manuscript).

Reviewer 2 Report

1、In Figure 1, I think it may be better to add legends or use different colors to represent different compounds.

2、Line 141-142: please explain which functional groups of levofloxacin are characteristic peaks at 3200-3500 cm-1 and 1700-1800 cm-1.

3、Line 206-233, Line 299-340: the paragraphs in this section are too long, and I think it would be better to break them down into several paragraphs to understand easier.

4、Line 349: the 4. discussion should be deleted.

5、In the part of conclusion, the author compares LVODT and LVT with various measurement methods, which I think should be pointed out in the part of conclusion. In addition, the significance of the article should also be mentioned, which I think is too simple in this part.

Moderate editing of English language required.

Author Response

1.  In Figure 1, I think it may be better to add legends or use different colors to represent different compounds.

Answer:  As you pointed out, I have colored both spectra to distinguish them (Fig. 1, L. 149-152 in revised manuscript).

2. Line 141-142: please explain which functional groups of levofloxacin are characteristic peaks at 3200-3500 cm-1 and 1700-1800 cm-1.

Answer: 3200-3500 cm-1:The peaks in this wavenumber range are thought to reflect the hydroxyl groups of the carboxylic acids and crystalline water.

1700-1800 cm-1:The peak in this wavenumber region is thought to originate from the carbonyl group of LV.

This information has been added to the text (L. 141-144 in revised manuscript).

3. Line 206-233, Line 299-340: the paragraphs in this section are too long, and I think it would be better to break them down into several paragraphs to understand easier.

Answer: We have divided the specified section into several paragraphs as pointed out (L. 208-243 and L. 302-344 in revised manuscript).

4. Line 349: the “4. discussion” should be deleted.

Answer: Thank you for pointing that out. We have deleted the phrase "4. Discussion" and have revised the “5. Conclusion" to "4. Conclusion" (L. 353 in revised manuscript).

5. In the part of conclusion, the author compares LVODT and LVT with various measurement methods, which I think should be pointed out in the part of conclusion. In addition, the significance of the article should also be mentioned, which I think is too simple in this part.

Answer: Other reviewers have pointed out that the conclusions are too brief. Following the reviewer's helpful advice, we have added the following text to this part (L. 354-366 in revised manuscript).

Reviewer 3 Report

In this work, the author evaluated the  levofloxacin in an orally disintegrating tablet, the author claimed to find a new usefulness of low frequency (LF) Raman spectroscopy. However, Raman technology is very mature in the direct imaging the distribution of drug molecules. The author used IR and Raman imaging to image the active compounds in the drug, but the experimental results obtained did not apply to optimization of the preparation process of the tablet, prescription process, and drug efficacy optimization research, so,the significance and value of this study need to be deliberated. After my careful consideration, the paper is not recommended to publish on Pharmaceutics.

Moderate editing of English language required

Author Response

Answer: 

Our research may not fit your research orientation (optimization of the preparation process of the tablet, prescription process, and drug efficacy optimization research). However, as shown in this study, understanding the special structure of OD tablets is useful for investigating the causes of defective products and is important for design and manufacturing process control. We believe that LF-Raman spectroscopy can be of great help here.